# Performance of Common Scene Stacking Atmospheric Correction on Nonlinear InSAR Deformation Retrieval

**Zhichao Zhang** [1,2], **Wanpeng Feng** [1,2,3,*] , **Xiaohua Xu** [4,5] and **Sergey Samsonov** [6]

1   School of Earth Sciences and Engineering, Sun Yat-sen University (Zhuhai), Zhuhai 519000, China; zhangzhch25@mail2.sysu.edu.cn
2   Southern Marine Science and Engineering Guangdong Laboratory (Zhuhai), Zhuhai 519000, China
3   State Key Laboratory of Earthquake Dynamics, Institute of Geology, China Earthquake Administration, Beijing 100029, China
4   University of Science and Technology of China, Hefei 230026, China; xiaohua-xu@ustc.edu.cn
5   Mengcheng National Geophysical Observatory, University of Science and Technology of China, Hefei 230026, China
6   Canada Centre for Mapping and Earth Observation, Natural Resources Canada, Ottawa, ON K1A0E4, Canada; sergey.samsonov@nrcan-rncan.gc.ca
*   Correspondence: fengwp5@mail.sysu.edu.cn

**Abstract:** Atmospheric Phase Screen (APS) is a major noise that suppresses the accuracy of InSAR deformation time series products. Several correction methods have been developed to perform APS reduction in the InSAR analysis, in which an algorithm called Common Scene Stacking (CSS) method draws wide attention in the community as the method was supposed to effectively separate atmospheric contributions without any external data. CSS was initially proposed for solving linearly interseismic deformation. Whether CSS can be applied in nonlinear deformation cases remains unsolved. In this study, we first conduct a series of data simulations including variable elastic deformation components and also propose an iterative strategy to address the inherent weak edge constraint issues in CSS under different deformation conditions. The results show that signal-to-noise ratio (SNR) is a key parameter affecting the performance of CSS in APS separation. For example, the recovery rate of deformation can generally be greater than 80% from datasets with SNR greater than 10 dB. Our results imply that CSS can favor further improvement of InSAR measurement accuracy. The proposed method in this study was applied to assessing deformation history across the 2020 Mw 5.7 Dingjie earthquake, in which logarithmic postseismic deformation history and coseismic contribution can be successfully retrieved once.

**Keywords:** Interferometric Synthetic Aperture Radar (InSAR); atmospheric correction; Common Scene Stacking (CSS); nonlinear deformation; InSAR time-series

## 1. Introduction

Interferometric Synthetic Aperture Radar (InSAR) has become a widely-used tool for monitoring surface deformation at centimeter to sub-centimeter levels in multiple earth sciences fields such as urban subsidence, landslides, volcanoes, earthquakes, and climate change related surface processes [1–8]. With the continuous accumulation of SAR data, particularly in a short revisit interval (≤12 days) like the Sentinel-1A/B SAR constellation [9], InSAR becomes increasingly practical, particularly in the assessment of potential geohazard risks. However, it is still challenging for InSAR to measure small deformations due to significant noise or errors propagated from the path of SAR signal travel [10,11], which can seriously undermine the accuracy of InSAR deformation products.

Atmospheric perturbations are among the primary noise sources in InSAR. The propagation of SAR signals through the ionosphere and troposphere can bring path information

into the SAR signals, and the estimation of this source is commonly known as Atmospheric Phase Screen (APS). Once the atmospheric conditions at two SAR acquisitions vary, atmospheric-related components will affect the resulting interferometric phase [12,13]. In comparison to tropospheric perturbations, ionospheric effects are more sensitive to long-wavelength SAR data (e.g., L-band SAR data). Generally, the ionospheric errors are sometimes neglected in mid-latitude regions for C-band interferograms [14]. Tropospheric phase delay is the major noise source for most InSAR applications. When the relative humidity of the troposphere changes by up to 20%, the atmospheric contributions in C-band interferograms may reach up to tens of centimeters [15]. Obviously, the magnitude of atmospheric noise could have been much larger than the target signals in numerous InSAR measurement scenarios [16]. Therefore, the effective reduction of atmospheric noise in InSAR applications is a crucial step.

Regarding the separate mechanisms of the tropospheric InSAR noise, previous effective atmospheric correction methods can be roughly grouped into two types [17–19]. One is based on external data to estimate the stratified and turbulent components, while the other is empirical and relies on the interferograms themselves to suppress the randomness in time due to tropospheric turbulence. The former can depend on the correlation between local elevation and interferometric phase [20,21] or utilize various weather data directly to model phase delays [22–27]. For example, Chaabane et al. [21] built up a linear model with elevation to correct interferograms for APS contributions at a global scale and showed a 54% reduction in the average uncertainty of the stacked deformation maps over Greece. In addition, the exact SAR signal delay from the troposphere can be modeled with atmospheric conditions provided by Global Navigation Satellite Systems (GNSS) [22], the Medium Resolution Imaging Spectrometer (MERIS) [23], the Moderate Resolution Imaging Spectroradiometer (MODIS) [24], and numerical weather models such as the European Centre for Medium-Range Weather Forecasts (ECMWF) Reanalysis v5 (ERA5) [27]. As a typical application, the Generic Atmospheric Correction Online Service (GACOS) relies on numerical atmospheric reanalysis datasets and regularly provides APS products globally. Their results showed consistent and even better performance with/than the global average GPS data and MODIS data [25,26]. However, based on previous studies, corrective effects are found to be limited to about 50% due to the challenge of modeling the turbulent tropospheric APS using external atmospheric or DEM datasets.

Several empirical algorithms have been proposed based on the tempo-spatial characteristics of the atmosphere itself, which typically assume a Gaussian distribution of atmospheric delay phases, including the stacking method [28,29], Persistent Scatterer (PS) [8,30], Small Baseline Subset (SBAS) [31–33], and the wavelet multiscale analysis method [34]. Evidently, this assumption may not be true in some cases. The strength of filtering is mainly based on the users' arbitrary choices. Therefore, it is hard to assess the performance of the APS reduction without additional observation, such as GNSS data.

The Common Scene Stacking (CSS) method is an algorithm to estimate APS components without external data requirements, initially proposed by Tymofyeyeva and Fialko [35]. The algorithm is fully based on the propagation characteristics of atmospheric components in interferograms sharing a common SAR acquisition with a linear deformation trend assumption. Tymofyeyeva and Fialko [35] first tested the algorithm using synthetic data and showed that 85–95% of the atmospheric signal could be separated. Tymofyeyeva et al. [36] further used the CSS method for interferograms covering the San Jacinto Fault (SJF) region with a APS reduction rate of about 67%. Xu et al. [37,38] applied the method to subsidence rate calculations at the Cerro Prieto Geothermal Field (CPGF), in which they revealed that the CSS approach could also seize phase discontinuities across adjacent bursts due to misalignment for Sentinel-1 interferometry. The potential of the CSS method is being recognized and ingested into time series analysis [39–41]. In practice, nonlinear deformation (e.g., a sudden surface change due to an earthquake or landslide) processes are common on Earth. Whether CSS can be applied

to handle more complex deformation processes with the fast growth of SAR acquisitions remains unanswered.

Theoretically, CSS estimates the APS component of a given SAR acquisition with a pair of interferograms involved in the scene as primary and secondary images, respectively. Then, those SAR data at the edges of the SAR series inherently cannot have enough constraints for their APS component estimation. How the APS components of those edge SAR acquisitions gradually propagate into deformation time series is not well understood. Regarding the increasing attention on CSS, an efficient way to enhance APS estimation for edge acquisitions is also highly demanded.

The aim of our study is to examine the performance of CSS in different deformation processes, e.g., inter-, co-, and/or postseismic deformation below a certain atmospheric level, and propose an iterative method to refine APS components of edge scenes in time series analysis. Finally, we apply our method to retrieve the deformation history of the 2020 Mw 5.7 Dingjie earthquake.

## 2. Methodology

### 2.1. Common Scene Stacking

The APS of an interferogram is the result of primary and secondary SAR images bringing the different atmospheric path delays into the interferometric phase. As shown in Figure 1, the green star indicates common SAR data that are used with other SARs (blue dots) to form interferograms. For example, three consequence SAR data, $i$, $i-1$, and $i+1$ (Figure 1), can form two interferometric phase pairs, $\Delta\phi_{i-1,i}$ and $\Delta\phi_{i,i+1}$, at any pixel. [35] simplified the two interferometric phases with APS ($\alpha_{i-1}$, $\alpha_i$ and $\alpha_{i+1}$) and linear trend deformation rate in two parts. Therefore, a subtraction of $\Delta\phi_{i-1,i}$ and $\Delta\phi_{i,i+1}$ can directly remove the deformation contribution and APS component $\alpha_i$ of date $i$ and can be calculated by Equation (1) (below):

$$\alpha_i = \frac{\Delta\phi_{i-1,i} - \Delta\phi_{i,i+1}}{2} + \frac{(\alpha_{i-1} + \alpha_{i+1})}{2} + \varepsilon \tag{1}$$

where $\alpha_i$ ($i > 1$) is the atmospheric delay phase of the shared date, $\Delta\phi$ is interferometric phase, and $\varepsilon$ represents noise.

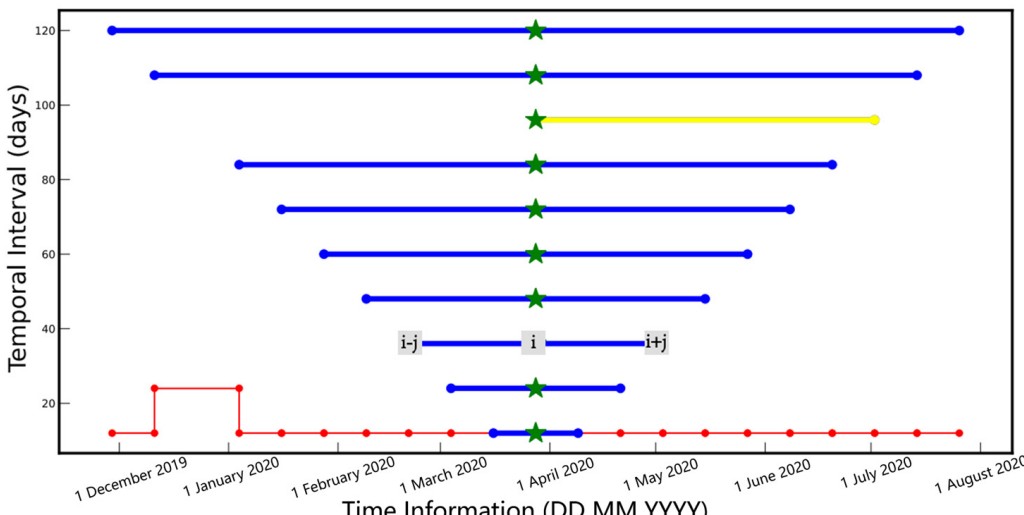

**Figure 1.** Interferogram network used in the CSS APS estimation. Blue dots represent the date of SAR acquisition, green pentagrams represent the common collection date, red dots represent the collection time interval, and the blue lines represent the interferogram pairs that are involved in CSS estimation, while the yellow ones are not.

By stacking all N interferogram pairs, the APS $\alpha_i$ at date $i$ is enhanced by 2 N times, while the other APS components are further weakened. When N is large enough in the

limit, it can preliminarily obtain the atmospheric phase $\alpha_i$ without needing to perform inversion as shown in Equation (2) [35]:

$$\alpha_i = \lim_{N \to \infty} \frac{1}{2N} \sum_{j=1}^{N} \Delta\phi_{i(i-j)} - \Delta\phi_{(i+j)i} = \lim_{N \to \infty} \frac{1}{2N} \sum_{j=0}^{N-1} (N-j) \left[ \Delta\phi_{(i-j)(i-j-1)} - \Delta\phi_{(i+j+1)(i+j)} \right] \tag{2}$$

The core step of the CSS method is to calculate the APS of individual SAR acquisitions in order based on the Atmospheric Noise Coefficient (ANC) (Equation (3)) during the iterative process, which means that the APS component with maximum APS effects should be calculated first and removed as well before estimating the APS for next acquisition.

$$ANC_i = (10.0)(R_{max})^{-1} \sqrt{\frac{1}{M} \sum_{m=1}^{M} (\alpha_i(x_m) - \overline{\alpha_i})^2} \tag{3}$$

where $R_{max}$ represents the RMS value of the APS with the maximum amount of noise.

Once the calculation is completed, the estimated atmospheric phases are subtracted from the interferogram, and then nearly "APS free" interferograms can be ready for deformation history with a simple SVD decomposition. Inherently, the edge SAR acquisition in the interferogram network cannot find two interferograms to form Equation (1), so in the early iteration, zero APS for the edge acquisitions is usually assumed [35]. Note that in later analysis, we perform traditional CSS APS separation with *sbas_parallel* released in GMTSAR [37].

### 2.2. Improved Time Series Analysis Method for nonlinear Deformation History

Time series InSAR techniques mostly focus on the deformation characteristics in the InSAR network to estimate deformation series, and they usually reduce noise sources in advance with temporal and/or spatial filtering. Among these algorithms, SBAS-InSAR is one of the most classic InSAR time series methods based on simple primary images, which only uses interferometric pairs with a short temporal baseline. The method organizes the small baseline differential interferograms as a linear model [31–33]:

$$A\phi = \delta\phi \tag{4}$$

In practice, the inversion of the InSAR phase for deformation (Equation (4)) is always an underdetermined issue in the presence of multiple interferogram subsets. This is why a deformation model (e.g., a linear or high-order deformation model) is required, as proposed by Berardino et al. [31]. In addition, a smoothing operator is sometimes suggested to be added to $A$ to reduce the rank deficiency [32]. To mitigate atmospheric effects in Equation (4), spatial and temporal filtering have been commonly used in time series applications [33].

Compared to the approaches above, CSS can help estimate APS for each SAR acquisition independently. The limited precision of APS at the edge of SAR acquisitions should also be considered amid the conditions where the signal of interest appears at the start or end of the time series (e.g., postseismic deformation). Here, we further extend coefficient matrix $A$ by involving APS components [42], in which a sudden deformation can also be considered. Then, in a case with N acquisitions and M interferograms, the signal contributions of the interferograms can be re-expressed as follows:

$$\boldsymbol{G}\,\boldsymbol{m} = \boldsymbol{d} \tag{5}$$

where $\boldsymbol{G}$ is the extended $A$, an (M + N − 2) × (N + 2) coefficient matrix, $\boldsymbol{d}$ is an (M + N − 2) × 1 matrix with M InSAR observations and N − 2 approximate APS solutions, and $\boldsymbol{m}$ are the components to be solved. Equation (5) can then be expanded in detail as follows:

$$\begin{pmatrix} \Delta t_1 & 1 & -1 & 0 & 0 & \cdots & 0 & 0 & 0 \\ \Delta t_2 & 1 & 0 & -1 & 0 & \cdots & 0 & 0 & 0 \\ \Delta t_3 & 1 & 0 & 0 & -1 & \cdots & 0 & 0 & 0 \\ \vdots & \vdots & \vdots & \vdots & \vdots & \ddots & \vdots & \vdots & \vdots \\ \Delta t_k & 0 & 0 & 0 & 0 & \cdots & 1 & -1 & 1 \\ 0 & 0 & 1 & 0 & 0 & \cdots & 0 & 0 & 0 \\ 0 & 0 & 0 & 1 & 0 & \cdots & 0 & 0 & 0 \\ 0 & 0 & 0 & 0 & 1 & \cdots & 0 & 0 & 0 \\ \vdots & \vdots & \vdots & \vdots & \vdots & \ddots & \vdots & \vdots & \vdots \\ 0 & 0 & 0 & 0 & 0 & \cdots & 1 & 0 & 0 \end{pmatrix} \begin{pmatrix} v \\ \alpha_{APS1} \\ \alpha_{APS2} \\ \alpha_{APS3} \\ \vdots \\ \alpha_{APSn} \\ C_{dis} \end{pmatrix} = \begin{pmatrix} \Delta\phi_1 \\ \Delta\phi_2 \\ \Delta\phi_3 \\ \vdots \\ \Delta\phi_k \\ \alpha'_{APS2} \\ \alpha'_{APS3} \\ \alpha'_{APS4} \\ \vdots \\ \alpha'_{APS(n-1)} \end{pmatrix} \quad (6)$$

where in $G$, $\Delta t_i$ is the time interval (in units of days) of the interferogram $\Delta\phi_i$ and 1 and $-1$ appear at the index of the primary and secondary dates [31], respectively, directly depending on the set of interference phases $\Delta\phi_i$. The last column in $G$ is related to the date of an earthquake. $v$ is the rate of linear deformation; $C_{dis}$ means coseismic displacement; $\alpha'_{APSi}$ is approximate solution of $i$th APS component estimated by the CSS method; and $\alpha_{APSi}$ is the re-estimated $i$th APS value. Obviously, the APS of edge SAR images can be further refined.

From Equation (6), we can further estimate potential co- and/or postseismic contributions. Their joint contributions can be modeled with the equation below [43]:

$$\phi(t) = H(t - t_0)\left[C + Kln\left(1 + \frac{t}{\tau}\right)\right] + Vt + b \quad (7)$$

where $\phi(t)$ is the surface deformation at time $t$; $H(*)$ is a Heaviside step function, $t_0$ is the date of a seismic event; $C$ is the coseismic displacement; $K$ is a constant; and $\tau$ is the decay coefficient (in units of days), representing how fast the postseismic transient decays with time. These two parameters are related to the deformation trend of the logarithmic function in postseismic; $V$ is interseismic linear deformation rate, and $b$ is a constant shift in observations. These five parameters of any station can be obtained with the nonlinear least squares method, for example, with the *curve_fit* function provided in *scipy* module of Python.

We first use the traditional CSS method to obtain an initial estimate of APS products. We then refine the APS contributions in an integrated solver (Equation (6)) using an iterative strategy. Then, the final APS is removed from the original InSAR network to obtain the updated InSAR dataset, that is expected to be nearly "APS free". Finally, we obtain the final LOS direction deformation sequence results through a simple SVD inversion.

Figure 2 illustrates the procedure of InSAR deformation history inversion with an iterative method, in which a refinement of APS components at edge SAR images has been considered. We name the procedure iCSS in the later section to be different from the traditional CSS method. As GACOS or ERA5 APS correction is SAR data based, it is confirmed that the even inaccurate APS components applied to the entire InSAR network would not destroy phase closures. If APS obtained through step 1 (Figure 2) is good enough by luck, the second term in Equation (1) should then be smaller in later analysis, implying that GACOS and ERA5 can increase the chance of successful application.

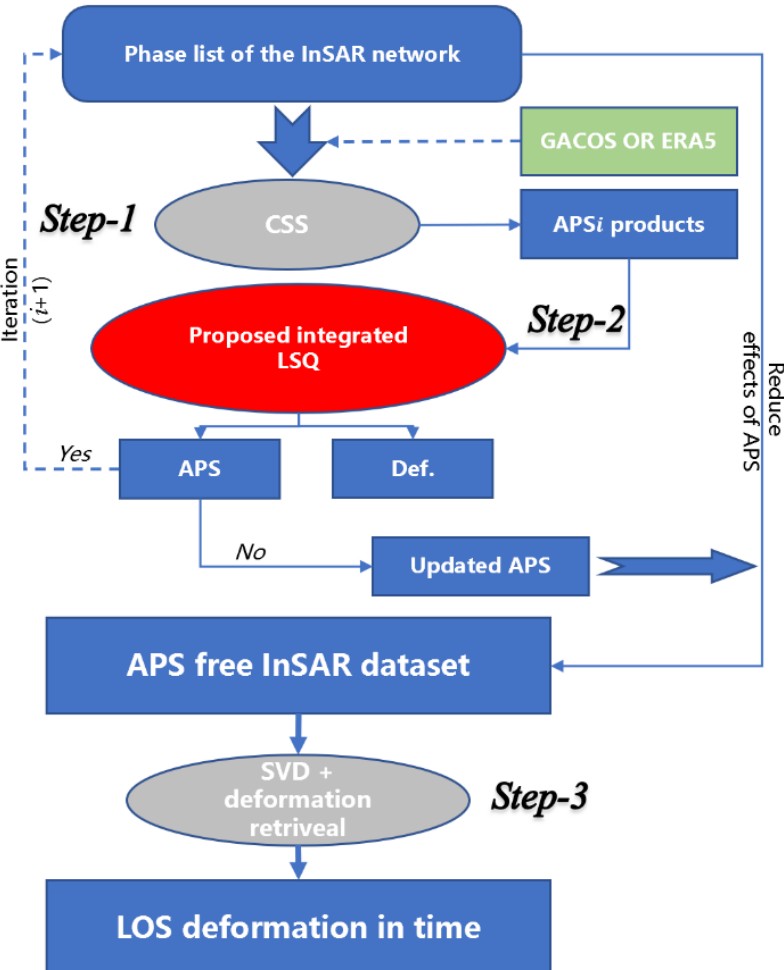

**Figure 2.** The flowchart of the InSAR time series analysis proposed in this study.

*2.3. Numerical Experiments*

2.3.1. Forward Modeling for the Earthquake Cycle

To examine the performance and validation of the proposed time series in this study, we construct a theoretical deformation time series that can consider the whole deformation cycle during an earthquake cycle. This includes a linear interseismic slip rate, a sudden change induced by the earthquake, and a logarithmic postseismic process (Figure 3). Their relative source depths have also been considered in the implementation of coding. A simple elastic half space dislocation, the Okada model [44], is adapted for all three deformation phases.

To speed up the analysis, we have developed a Python script that allows a few parameters to automatically create three-dimensional (3D) deformation for a region and form 1D LOS InSAR data based on the given time and SAR geometries. In simulation, spatially correlated atmospheric contributions are considered [45,46], and different noise levels can be applied. Then iCSS (Figure 2) is performed to separate the deformation history and APS components of the simulated dataset, respectively. Note that we also apply iCSS to the 2020 Mw5.7 southern Tibet earthquake. So, all the synthetic data in the following session are consistent with the true SAR data used for the 2020 earthquake in both time and space.

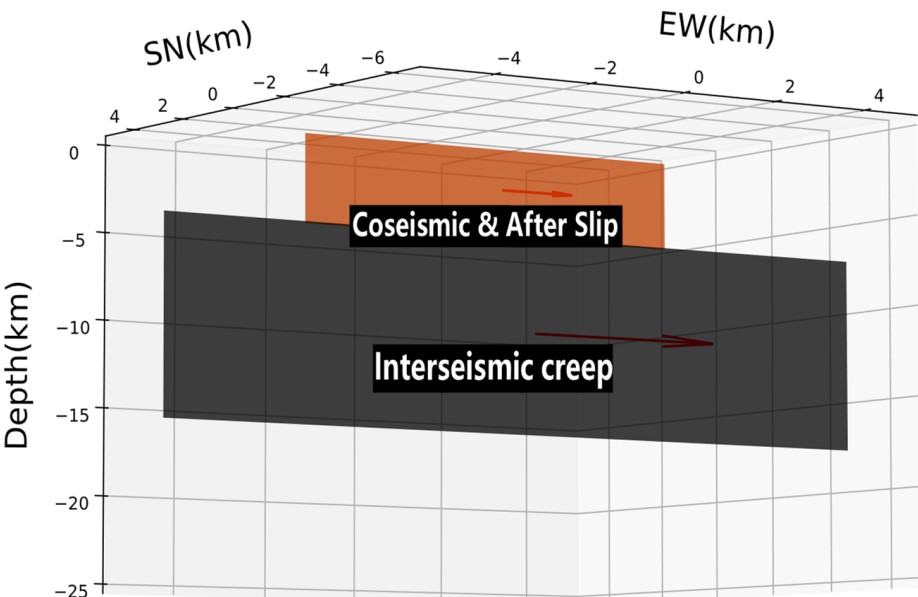

**Figure 3.** Fault slip models for theoretical deformation simulation.

### 2.3.2. Performance Evaluation of the Inversion Results

In the simulation, we export the deformation and APS (random noise) components, respectively, as truth values for comparison. We estimate APS and deformation starting from InSAR synthetic datasets. We use the root mean square error (RMSE) of residual APS as an indicator to evaluate the performance of iCSS for APS. We also adopt the recovery rate of deformation and APS components, respectively, as indicators of correction efficiency, which is defined as follows:

$$recovery\ rate = \left(1 - \frac{|Val_{input} - Val_{output}|}{|Val_{input}|}\right) \times 100\% \tag{8}$$

where $Val_{input}$ indicates the theoretical value of the input, and $Val_{output}$ is obtained from the inversion based on iCSS.

In addition, we analyze the relationship between atmospheric noise level and recovery rate, as well as the relationship between the signal-to-noise ratio (SNR) of input data and recovery rate at different levels of atmospheric random noise. SNR is defined in dB as follows [47]:

$$SNR = 10 \times \log_{10}\left(\frac{P_{signal}}{P_{noise}}\right) \tag{9}$$

where $P_{signal}$ indicates power of the deformed signal, and $P_{noise}$ indicates the power of APS noise. Here, power is calculated by averaging the sum of the squares of the signal.

### 2.3.3. D-InSAR Processing

In this study, we perform D-InSAR processing of Sentinel-1 TOPS-SAR data using an InSAR automated processing environment, *pSAR* [48,49], in which the kernel is the open-source InSAR processing system, the Generic Mapping Tools (GMTSAR6.2) [50]. We use the Precise Orbital (PREORB) data and external 30-m resolution Shuttle Radar Topography Mission (SRTM) DEM data [51] in the InSAR processing for coregistration and topography correction. In the processing, the date we have chosen for the master image is 28 March 2020. We set a temporal baseline threshold of 120 days to conduct interferometric pairs to ensure an adequate number of interferograms while maintaining good coherence, and those pairs with SAR acquisitions in similar seasons, but in different years have also been processed. All interferograms are multilooked with look numbers 8 and 2 in range and azimuth directions, respectively, and filtered using the internal smoothing strategy of GMTSAR, including

Gaussian and Goldstein smoothing methods [52], to reduce noise level. Then we unwrap the phases based on the statistical-cost, network-flow algorithm (SNAPHU) [53] with a coherence threshold of 0.15, taking into account the excellent interferometric coherence in our study area. Finally, the interferograms are all geocoded to the WGS84 projection with longitude and latitude coordinates for later analysis.

## 3. Results

### 3.1. Validation with a Synthetic Dataset

We first simulate InSAR surface time series, including deformation and random APS noise, with various deformation mechanisms (Supplementary Materials Figures S1, S2, and Figure 4). To examine the performance of CSS, we generate multiple APS datasets with increasing noise levels for each deformation case. Figure 4 is based on a complex deformation model. In the input data period from 2016 to 2022, including 203 SAR acquisitions and 4270 interferograms, an earthquake is supposed to occur on 20 March of 2020, which has four clear coseismic deformation lobes with a maximum deformation of ~2 cm. A logarithmic postseismic deformation is followed immediately. For comparison, a traditional SBAS is also performed (referred to as SBAS2003) [32], in which the deformation results of iCSS are further applied to obtain APS components from the original dataset.

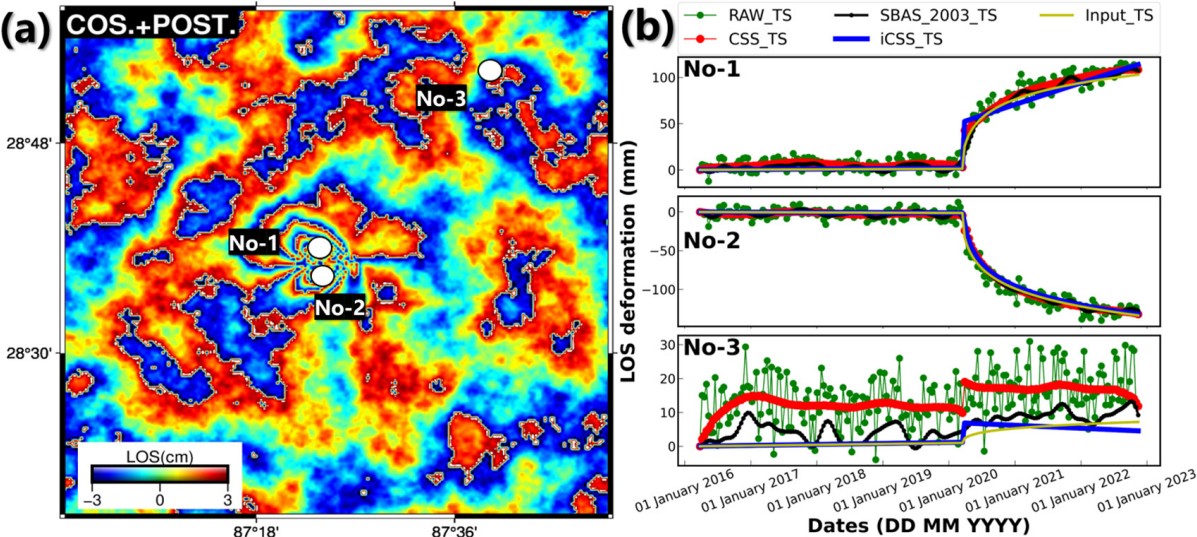

**Figure 4.** (**a**) An accumulated simulated interferogram of 9 February 2020–6 March 2022 with 20 mm of random noise added that includes co- and 2-year-long postseismic deformation, where the dots (white) are reference stations for time series analysis. (**b**) Deformation sequences obtained from inversion at the three stations, respectively.

Three stations (Figure 4) are selected for time series analysis. The results of No-1 and No-2 reveal clear and distinct cos- and postseismic trends, demonstrating that the CSS method is also valid for nonlinear deformation, including co- and postseismic deformation. At station No-3, it indicates that both conventional CSS and SBAS2003 can effectively suppress APS random noise, but the effectiveness of these correction methods is limited. In particular, the results of the deformation series at the far field point demonstrate the weak edge constraint of the CSS method. By using iCSS, the deformation sequences are better smoothed.

Through numerical tests with different noise levels, we find that the RMSE of APS estimated by iCSS remains low for all testing datasets with different deformation mechanisms (Table 1), which shows the effectiveness and availability of our algorithm. However, it is also clear that the deformation recovery rates decrease with increasing noise levels (Table 1), in which coseismic components can only be recovered (39.6%) at a noise level of 50 mm. The linear deformation inversion shows similar patterns, implying that iCSS is

good for APS estimation, particularly for the trends of time series. Once the deformation signal is weak (low SNR), the retrieved APS may still have high coherence with the input APS, but the deformation recovery rate can be very low.

**Table 1.** Performance of the iCSS method in different deformation experiments with variable APS magnitudes.

| Station Index | Noise Level (mm) | Deformation Model # | APS RMSE (mm) | Recovery Rate (Cos.) (%) |
|---|---|---|---|---|
| 1 | 10 mm | Linear | 0.91 | / |
| | | Cos. | 0.44 | 95.8 |
| | | Cos. + Post. | 4.20 | 50.4 |
| 2 | 20 mm | Linear | 0.38 | / |
| | | Cos. | 1.39 | 95.1 |
| | | Cos. + Post. | 2.82 | 74.4 |
| 3 | 50 mm | Linear | 4.07 | / |
| | | Cos. | 1.36 | 39.6 |
| | | Cos. + Post. | 5.23 | 0 |

Note: #: Three deformation mechanisms are considered in the simulation, in which "Linear" represents the interseismic process, "Cos." means a sudden deformation due to an earthquake, and "Post." means a logarithmic postseismic deformation process.

Figure 5 shows the relationship between random noise and recovery rate, SNR, and recovery rate, respectively. When random noise is at a low level (<4 cm), the recovery rate of deformation is up to more than 90%. Even in the case of low SNR, the recovery rate of our method can reach more than 70%.

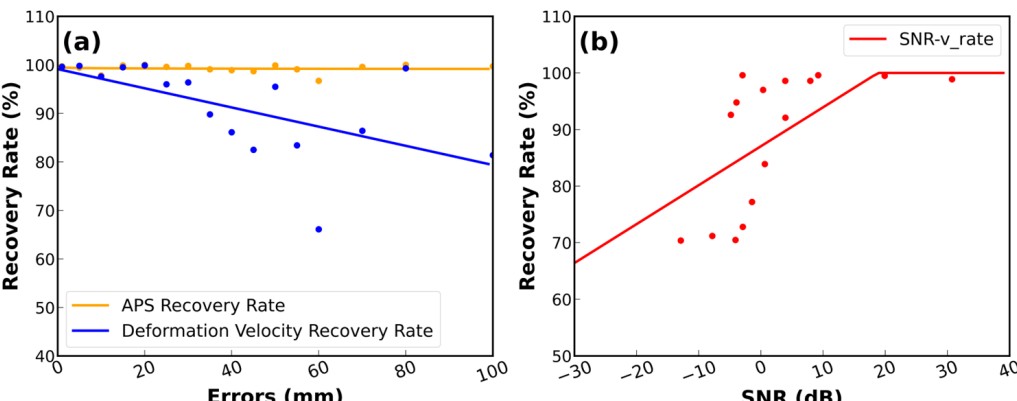

**Figure 5.** (**a**) Relationship between APS errors and recovery rate. The blue line represents the recovery rate of deformation velocity; the yellow line represents the APS recovery rate. (**b**) Relationship between SNR and recovery rate of deformation velocity.

To further test the feasibility of our proposed method, we simulate a shallow creep-slip process on the fault using synthetic data, which has the addition of 20 mm of random noise. The results along profile AA' retrieved using iCSS (Figure 6a) show that an average RMSE of residual average deformation rate at all reference points is 1.77 mm/yr and the absolute difference between iCSS and input deformation rates can be seen to be far smaller than the error bars given in the simulation, demonstrating the high correction efficiency of iCSS.

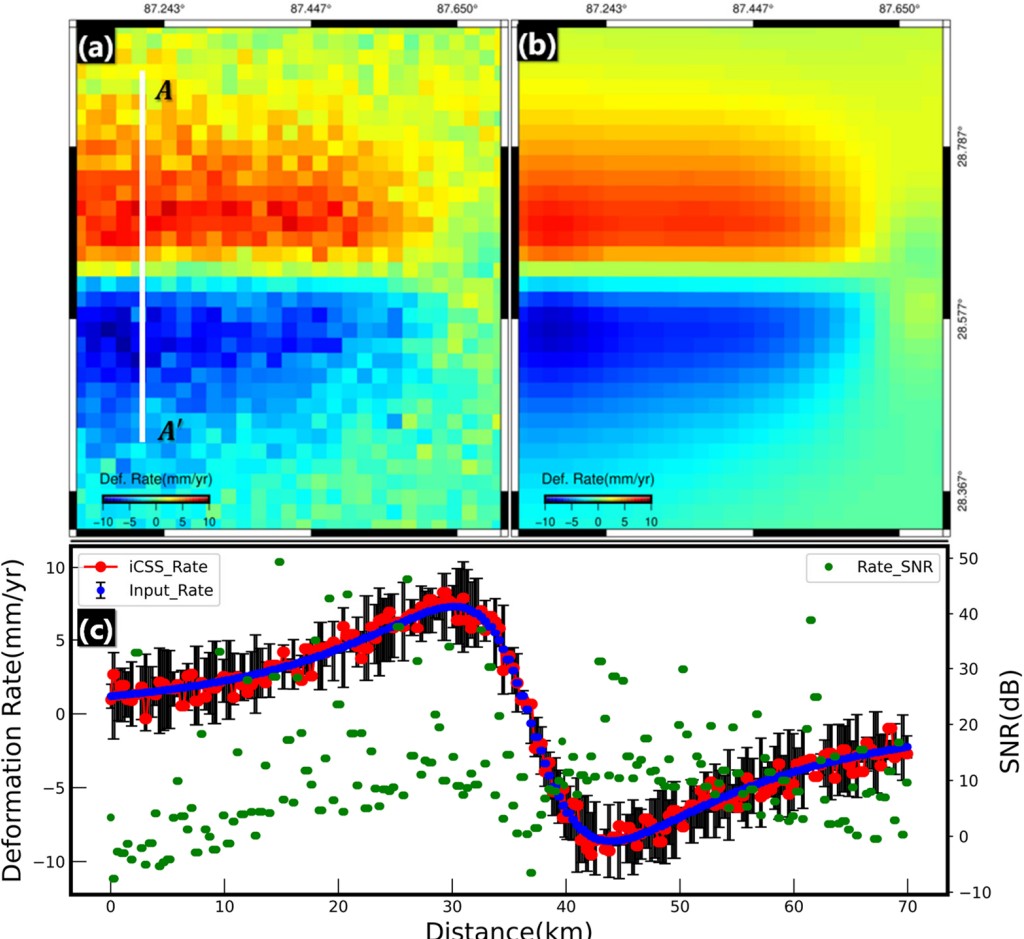

**Figure 6.** (**a**) Average velocity map inverted by iCSS. AA′ is the velocity profile across the linear deformation fault; (**b**) the input theoretical mean velocity map; (**c**) the average velocity along the cross-fault profile and its SNR distribution.

### 3.2. A Case Study on the 2020 Mw 5.7 Dingjie Earthquake

To map surface deformation history following the 20 March 2020 Mw 5.7 earthquake [54], we use the D-InSAR processing flow described in Section 2.3.3 to process Sentinel-1 TOPS SAR data in descending Track 121, including 186 acquisitions from 13 March 2016 to 25 November 2022. One swath of T121 fully covers the earthquake area, which is analyzed in the study (Figure 7). Finally, a total of 3578 interferograms are obtained, of which 1029 postseismic interferograms are included (Figure 8).

We apply iCSS for estimating surface deformation history. As iCSS is designed based on individual pixels and the green functions of every pixel need to be calculated separately, this makes the package time consuming. To reduce the processing time, we perform quadtree down-sampling [55] of the deformation field to extract limited points for time series estimation. Through the down-sampling processes, we reduce millions of points to ~2400 pixels. In some cases, the strong atmospheric signals may also lead to additional sampling in practice. So a model resolution based (Rb) sampling [46] with a fault geometry can also be considered instead to guarantee all sampled points are concentrated around the fault trace.

Before down-sampling, we use ERA5 model-based APS prediction to reduce APS effects in the original InSAR datasets [27,39]. For residual long-wavelength signals (mainly orbital errors) in the interferograms, we apply a three-parameter linear best-fitting plane for reducing those parts after removing the epicentral area. As we mainly care about co- and postseismic deformation, which are thought to be concentrated in the vicinity of the earthquake fault, there is no harm in removing the long-wavelet signals ahead.

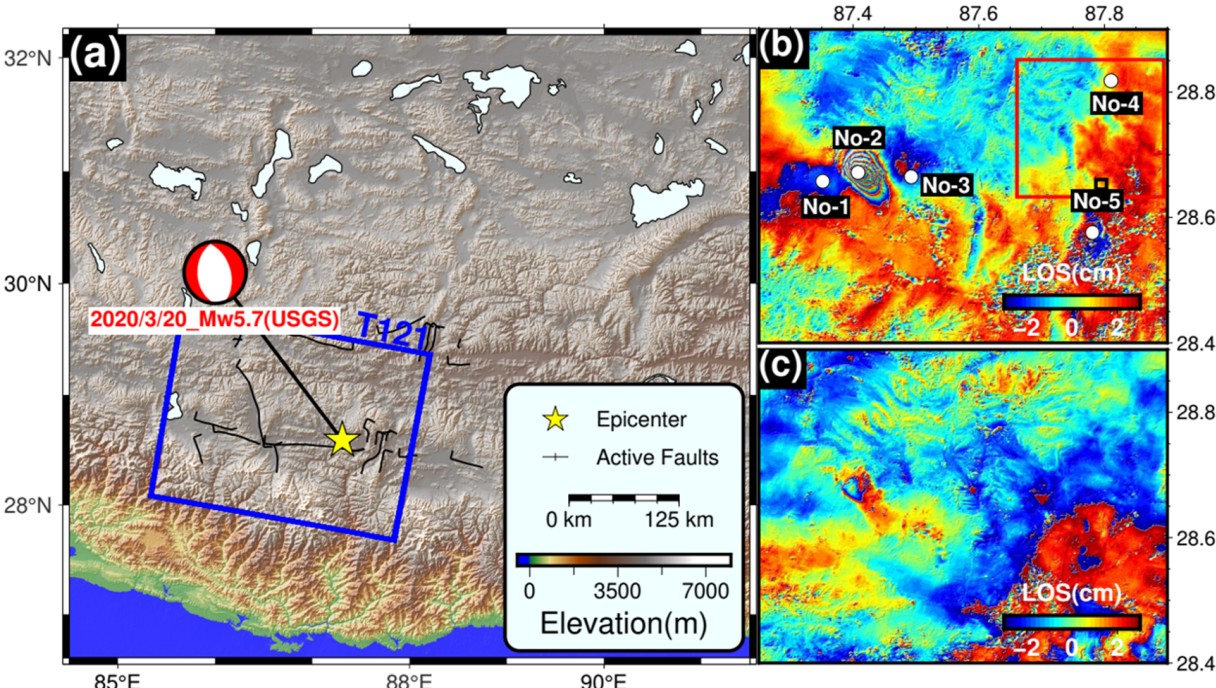

**Figure 7.** (**a**) Tectonic background of the study area. The blue rectangle is the area covered by Sentinel-1A descending track T121. (**b**) A coseismic interferogram of 3 March 2018–27 February 2021. White stations are selected for deformation time series analysis in later sections, in which No-1, No-2, and No-3 are located in the areas with significant deformation from the earthquake, while No-4 and No-5 are far from the epicentral area thought to be stable in time. The black box near No-5 is the reference area we selected for the system shift. The red box represents the subregions selected for full-resolution inversion. (**c**) A postseismic interferogram with clear near-fault deformation. All interferograms are rewrapped in a range from −3 to 3 cm, and the obvious atmospheric effects can be seen in both interferograms.

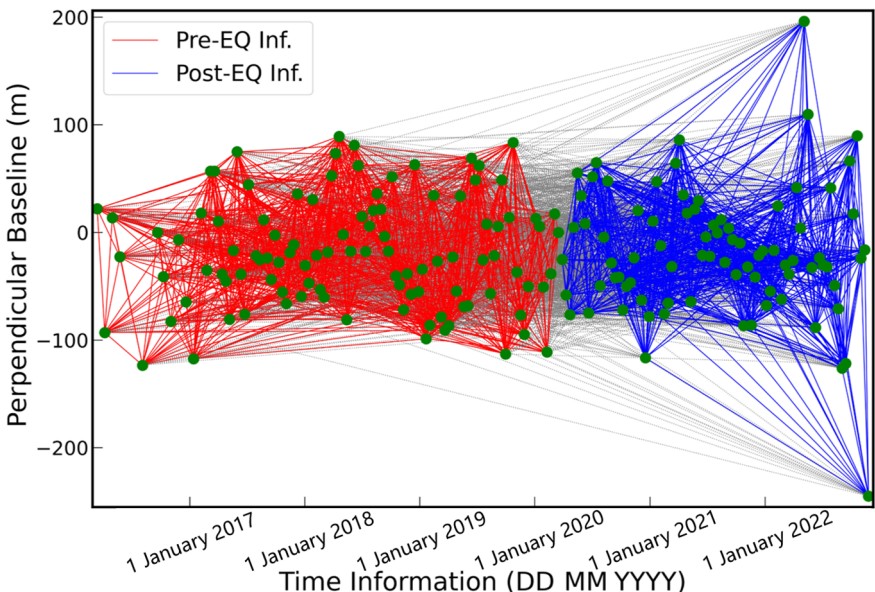

**Figure 8.** Temporal-spatial baseline plot of the processed interferometric pairs based on T121 Sentinel-1 datasets. Green dot: SAR acquisition date corresponding to the SAR data involved in interferometric processing; the red line indicates the interferograms consisting of SAR data acquired before the earthquake; the blue line indicates the interferograms consisting of SAR data acquired after the event; the grey line indicates the interferograms containing the coseismic deformation.

As shown in Figure 7, the topography in the study area is rough, and the mean elevation is ~4500 m. Three deformation centers can be identified directly in the coseismic interferogram (Figure 7b), while the accumulated postseismic deformation can also be spotted, corresponding to the maximum coseismic deformation, which is clearly less than the regional APS level of ~2 cm.

The inversion results of Stations 2 and 3, located both in the hanging wall, with one right above the slip center and the other a bit far to the east, respectively, reveal clear deformation history (Figure 9). No-2 has a clear sudden subsidence of ~125 mm and continues moving downward logarithmically with an accumulated displacement of ~30 mm in the postseismic period, while No-3 has a clear uplift trend of ~20 mm across the mainshock, but immediately moves inversely with an accumulated subsidence of 7 mm in the postseismic period. By contrast, No-4 is far from the epicentral area, with negligible deformation. Relative to the original time series, the correct deformation history shows a significant noise reduction from 20 mm to 2 mm level. This implies that the proposed strategy in this study has the potential to retrieve complex deformation histories.

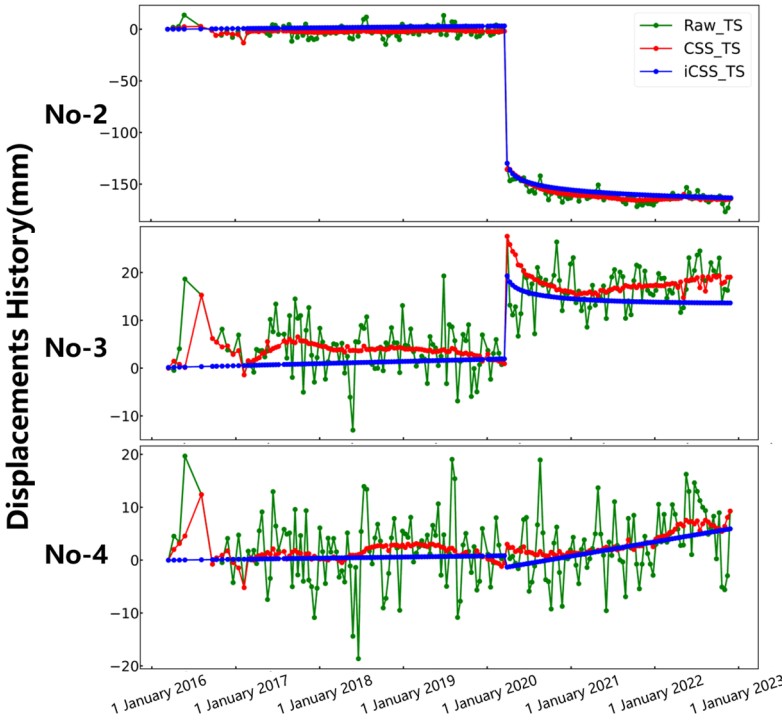

**Figure 9.** Deformation history estimated with iCSS. The green lines represent deformation sequences obtained by decomposition of raw data; the red lines represent deformation series results after CSS correction; and the blue lines are deformation sequences using iCSS.

The results indicate that iCSS can obtain a complete deformation sequence for an earthquake cycle. Meanwhile, the far field remains relatively stable, indicating that noise has been effectively removed. Our proposed method surpasses traditional InSAR time series analysis in producing such pristine deformation series; few existing InSAR time series methods can obtain such clean deformation results.

Meanwhile, a postseismic deformation time series has also been estimated, which can then be applied to compute a decay time $\tau$ with an internal nonlinear regression method in Python, e.g., *curve_fit*. $\tau$ is thought to be related to the friction coefficient of a fault [56,57]. Supplementary Materials Figure S3b shows that the $\tau$ in the epicentral area tends to be nearly constant, implying homogeneous fault physical properties in space. Note that the $\tau$ estimation may have large uncertainties, particularly for low deformation areas, which is due to non-uniqueness of nonlinear inversion.

To further demonstrate the superiority and effectiveness of our proposed method, we apply ERA5, GACOS, CSS, and iCSS to full-resolution interferograms for atmospheric correction. In order to better evaluate the correction results and to speed up the inversion, we selected only 1418 interferograms composed of 104 dates before the earthquake for a small area far from the epicenter (the red box in Figure 7b).

Figure 10 shows the APS components estimated from the different methods for the interferogram of 13 March 2016–27 March 2018, in which 13 March 2016 is an edge acquisition in the SAR dataset. Topographic-correlated APS can be found across the interferogram (Figure 10a,b). The APS obtained from both ERA5 and GACOS are almost identical (Figure 10c1–d1), while the APS components from CSS present conspicuous anomalies (Figure 10e1). Obviously, the APS retrieved by iCSS (Figure 10f2) shows the best APS correlation rate, with a SD of 0.221 cm for the residuals.

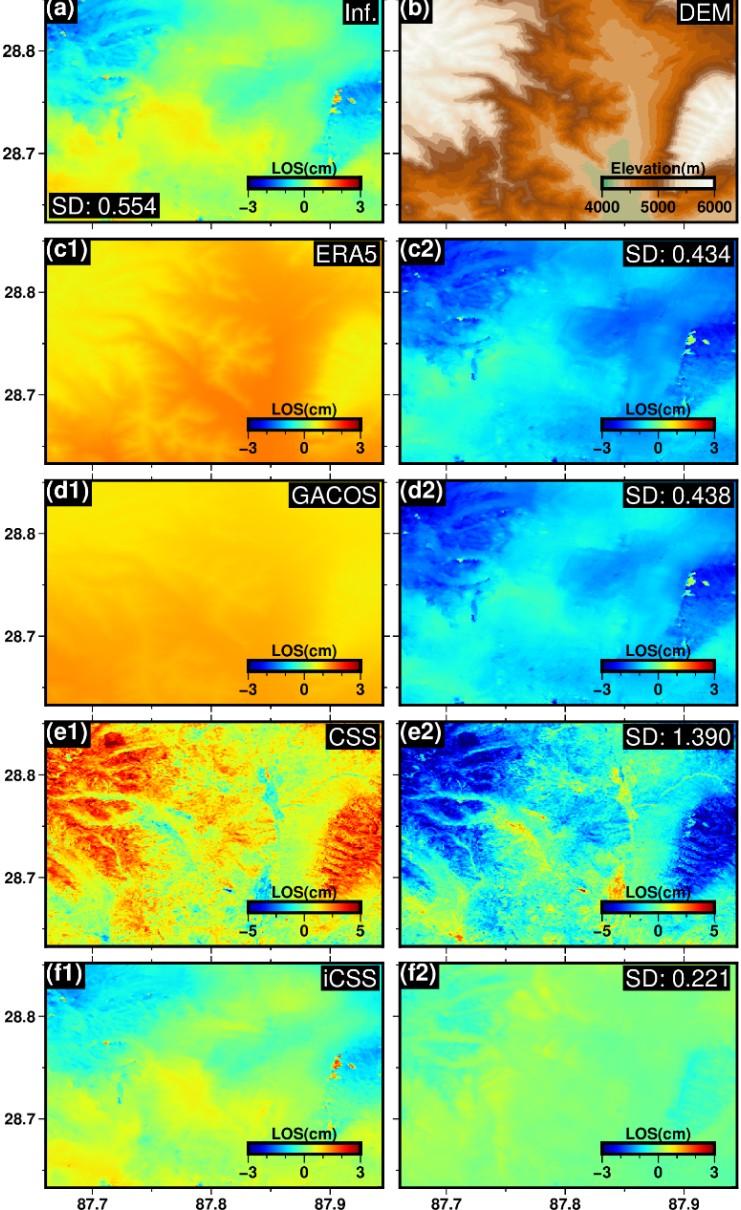

**Figure 10.** Comparison of the results of different methods for estimating atmospheric components. (**a**) Original interferogram of 13 March 2016-27 March 2018. Here, 'Inf.' is denoted as interferograms. (**b**) DEM map of the study area with an elevation difference of more than 2000 m. (**c1**) Atmospheric components are estimated based on ERA5. (**d1–f1**) as (**c1**) but for GACOS, CSS, and iCSS, respectively. (**c2–f2**) are (**c1–f1**) corresponding corrected interferograms.

Figure 11 shows similar trends in APS correction rates for 18 interferograms, which all involve 13 March 2016 as the primary SAR. It is clear that iCSS can nicely handle edge issues for all interferograms (all red bars in Figure 11). The performance of GACOS and ERA5 is not stable. This is likely due to the low resolution of the local atmospheric products. Note that the most right three pairs are made with seasonal data that have similar atmospheric conditions. Their original APS components were initially low.

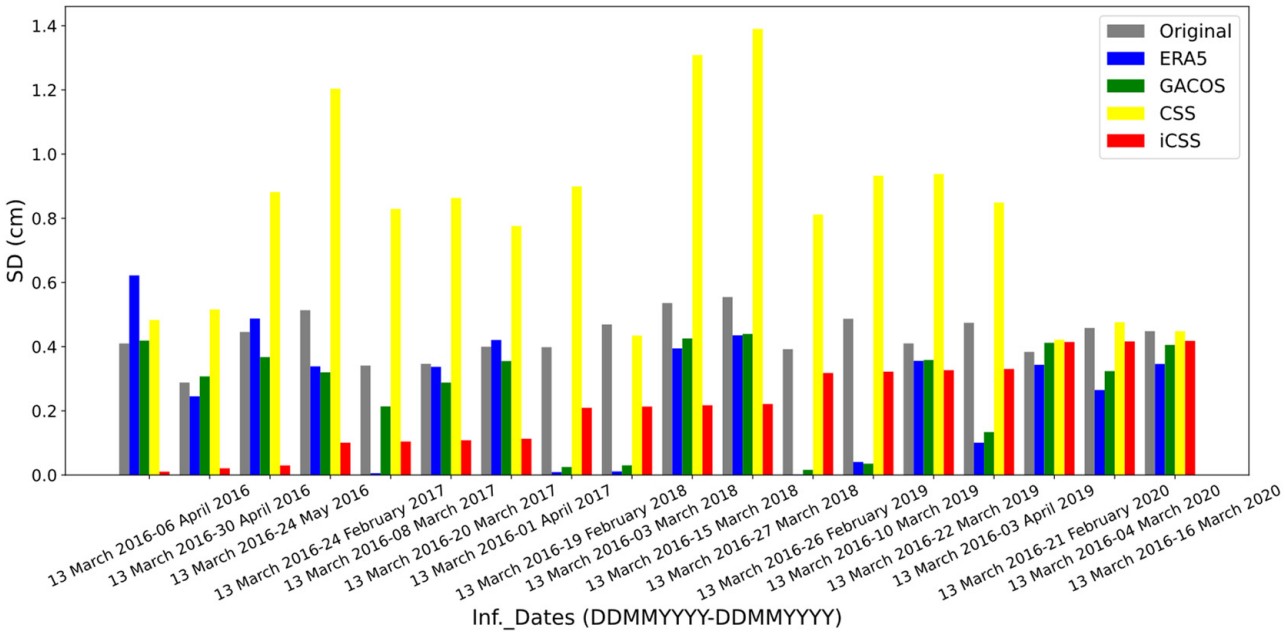

**Figure 11.** SD for all interferograms containing edge dates (13 March 2016). Bars with different colors indicate the SD results of the corresponding interferograms after atmospheric correction by different methods. Here, 'Inf.' is denoted as interferograms.

## 4. Discussion

Edge SAR data in the SAR network are clear weakness in CSS APS estimation. However, if the edge SAR data in the network inherently have limited APS effects, the potential issue with the edge SAR APS issues should be gone. As shown in Figure 12, we extract the APS component values from the ERA5 simulations at the five stations (black lines) and compare them with the APS obtained from the CSS inversion based on the different InSAR subsets.

As shown in Figure 12, the gray lines at stations 3, 4, and 5 show generally identical trends with ERA-5 predicted APS components, with a SD of 2.811, 2.279, and 2.603 between them (Table 2), which is lower than any of the other tests (red and blue lines). This also indicates that ERA5 APS models are fine for simulating APS components for use in the 2020 earthquake area in the selected period. However, variable correlations between ERA5 and CSS APSs for the 5 stations can be found, which is reasonable as the ERA5 datasets usually show different uncertainties in space [25].

APS seasonal trends can be identified at the 5 stations (yellow lines). It is clear that the APS components are smaller in winter (November–March) and larger in summer (June–September), up to more than 10 mm, which is consistent with the conclusions of previous studies [27]. Since there are weak constraints on the edge SAR APS components from the CSS method, we can intentionally choose the SAR with a small APS component as edge SAR data to improve the accuracy of APS estimation (e.g., SDs calculated for blue lines at Stations 3 and 5 are smaller than red lines). Therefore, for the selection of interferogram pairs, we need to consider the seasonal characteristics of the APS and select winter SAR data as the edge. Especially in the case of only linear deformation, image pairs with little atmospheric influence can be selected for interferometry.

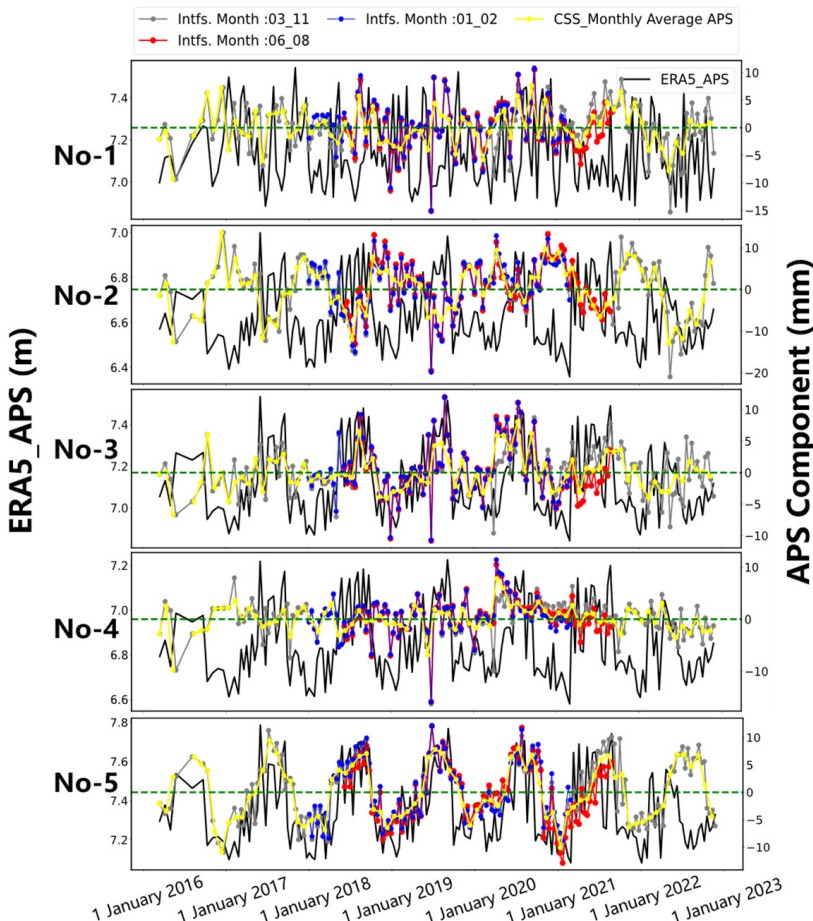

**Figure 12.** Analysis of atmospheric seasonal characteristics. The black lines represent the atmospheric phase delay estimated based on ERA5, while gray, blue, and red lines represent the APS component series extracted from edge SAR data with different seasons. This includes using the full dataset (gray), using summer (June–August) SAR data as edge data (red), and using winter (December–February) SAR data as edge data (blue). Yellow lines represent the monthly average APS derived from CSS with all data. Here 'Intfs.' is denoted as interferograms.

**Table 2.** Quantitative analysis of atmospheric seasonal characteristics.

| Station Index | Elevation (m) | Data Period | Difference * | Month of APS Component Maximum | Month of APS Component Minimum |
|---|---|---|---|---|---|
| 1 | 4795 | March 2016–November 2022 | 3.026 | 07 | 03 |
|   |      | June 2018–August 2021 | 2.856 | 09 | 03 |
|   |      | January 2018–February 2021 | 2.876 | 09 | 03 |
| 2 | 5299 | March 2016–November 2022 | 3.936 | 07 | 04 |
|   |      | June 2018–August 2021 | 3.953 | 12 | 05 |
|   |      | January 2018–February 2021 | 3.881 | 07 | 03 |
| 3 | 4730 | March 2016–November 2022 | 2.811 | 07 | 04 |
|   |      | June 2018–August 2021 | 2.996 | 07 | 03 |
|   |      | January 2018–February 2021 | 2.968 | 07 | 03 |
| 4 | 4850 | March 2016–November 2022 | 2.279 | 06 | 01 |
|   |      | June 2018–August 2021 | 2.563 | 04 | 02 |
|   |      | January 2018–February 2021 | 2.637 | 04 | 03 |
| 5 | 4307 | March 2016–November 2022 | 2.603 | 07 | 04 |
|   |      | June 2018–August 2021 | 2.961 | 07 | 05 |
|   |      | January 2018–February 2021 | 2.642 | 07 | 05 |

Here, '*' indicates the standard deviation of the differences between ERA- and CSS-based APS components.

In addition, the seasonal performance of different stations is inconsistent, which may be related to the geomorphology and elevation. The elevation of Station 2 is nearly 5300 m, which is usually related to terrain, so its seasonal characteristics are greatly affected by temperature. While the elevations of stations 3 and 5 are low (~4700 m and ~4300 m, respectively) and have little correlation with terrain, their seasonal characteristics are similar to the trend estimated by ERA5.

Meanwhile, as seen in the theoretical test (Figure 4) and real application (Figure 9), a slight overestimate of coseismic deformation for the far field points can be found. This seems inevitable in the proposed method in this study. As shown in Equation (6), the APS and coseismic deformation will be estimated through an entire linear form at the same time, implying that they will have a trade-off between each other. In the future, a sophisticated weighting operation can be considered to allow those non-deformation pairs to be fitted with top priority.

### 5. Conclusions

Correcting interferograms for APS components of the troposphere is an important step in a successful InSAR application, particularly for the study of small deformation. The traditional CSS algorithm, which assumes linear deformation and equal time baselines, has been widely used for separating atmospheric phases based on the data itself. However, this method suffers from weak edge constraint issues, and its applicability for non-linear or sudden deformation problems in practical applications is uncertain. In our study, we proposed an algorithm that can directly extract co- and postseismic deformations from complex deformation processes. We tested the feasibility and effectiveness of the algorithm with synthetic data and applied it to the 2020 Mw 5.7 Dingjie earthquake. Our findings suggest that:

1.  The CSS method does perform excellently for APS retrieval with abundant interferograms, which has been validated with the ERA5 APS simulation for application in southern Tibet.
2.  With the iterative way proposed in this study, iCSS can also be applicable for the estimation of cos- and/or postseismic deformation in time series analysis.
3.  Our proposed method (iCSS) has effectively addressed the weak APS constraint issue for edge SAR acquisitions in an iterative strategy in practical applications.
4.  For regions showing seasonal APS distributions, we suggest intentionally choosing SAR data with low APS effects as edge SAR data in the InSAR network.

**Supplementary Materials:** The following supporting information can be downloaded at: https://www.mdpi.com/article/10.3390/rs15225399/s1. Figure S1: An accumulated simulated interferogram of 9 February 2020–6 March 2022 with 20 mm of random noise added that includes linear deformation. Figure S2: An accumulated simulated interferogram of 9 February 2020–6 March 2022 with 20 mm of random noise added that includes coseismic deformation. Figure S3: The $\tau$ value result from inversion.

**Author Contributions:** Conceptualization, W.F.; visualization, Z.Z., W.F. and X.X.; writing—original draft preparation, Z.Z.; and writing—review and editing, W.F., X.X. and S.S. W.F. supervised the study and secured the funding. All authors have read and agreed to the published version of the manuscript.

**Funding:** This work was supported in part by the State Key Laboratory of Earthquake Dynamics (LED2021206), an open grant from the Lhasa National Geophysical Observation and Research Station (NORSLS20-02), and the National Natural Science Foundation of China (NSFC) (42274064).

**Data Availability Statement:** The Sentinel-1 SAR dataset and the precise orbit information were provided by the European Space Agency (ESA) through ESA's Scientific Sentinel-1 Hub. The ERA5 data were provided by ECMWF. A Python-based deformation simulation package is available on Github (https://github.com/wpfeng/pSAR_defsim, accessed on 24 September 2023).

**Acknowledgments:** We sincerely thank the anonymous reviewers for their insightful and constructive comments to improve the quality of this paper. Most figures were prepared using the public domain Generic Mapping Tools (GMT6.4.0) [50] and Python3.9.13 [58].

**Conflicts of Interest:** The authors declare no conflict of interest.

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
