# Peer review of "Performance of Common Scene Stacking Atmospheric Correction on Nonlinear InSAR Deformation Retrieval"

_remotesensing, doi:10.3390/rs15225399_

Round 1

Reviewer 1 Report

Comments and Suggestions for Authors

The CSS method suffers from weak edge constraint issues and its applicability for non-linear or sudden deformation problems in practical applications is uncertain. In this manuscript, an algorithm that can directly extract co- and post-seismic deformations from complex deformation processes is proposed. My main comments are as follows:

1.     In Introduction, the novelty/contribution in the Introduction is not clearly stated. The iterative approach proposed to address the weak edge constraint issues is compared to the traditional CSS method, but the improvement is not quantitatively described, the manuscript lacks quantitative comparisons.

2.     Regarding the real SAR data, the manuscript only provides time series deformation comparisons for three slip center points to validate its effectiveness. It suggested the results to be compared to that of GACOS/ERA5 or other more effective methods. Meanwhile, the interferograms before and after employing the proposed and comparison methods should be given to demonstrate the effectiveness of the proposed method.

3.     Discussion section discusses the selection of edge SAR data and obtained the following conclusion: ‘Since the weak constraints on the edge SAR APS components from the CSS method, we can intentionally choose the SAR with small APS component as edge SAR data to improve the accuracy of APS estimation’. However, as the new iterative method has been proposed to address the weak edge constraint issues, it should discuss/validate the performance of the proposed method on edge SAR data from different time periods. Thus, to validate the effectiveness and innovation of the proposed method.

4.     Further evaluation is required for the filtering or overestimate of deformation signals by the proposed method.

5.     Figure 8 missing unit in Y axis. Quality of the presented figures need to be improved. Other formatting problems required to be checked through the whole manuscript.

Comments on the Quality of English Language

No comments.

Reviewer 2 Report

Comments and Suggestions for Authors

The idea of this study is interesting, and the manuscript is well structured. However, the applicability of this method to full-resolution interferograms needs to be well examined and presented clearly. The down-sampling of interferogram resolution would average atmospheric effects. Estimating APS in full resolution is completely different from doing that for around 2400 large pixels. Reducing the number of images in the real showcase would be an option.

-        The authors need to specify the applicability of this method to different InSAR methods, not SBAS only.

-        The discussion needs to be improved. Some comparisons between this method's results and other related methods are essential.

-        Figures’ parts need to be labeled for better illustration and caption understanding.

-        Is the super primary image mentioned in figure 8 referring to the master image?

-        In Figure 7, the interferogram and labeled points are more important than the study area box. Consider switching the location and size of sub-figures a and b. Figure 7c also needs to be bigger to show the details.

-        Some technical writing issues exist in the manuscript. For instance:

“we apply our method to apply for retrieving deformation history across the 2020 Mw 5.7 Dingjie earthquake” at the end of the introduction.

“spatial and temporal filtering have been commonly used intime-series applications [34]” on page 4.

“We the refine the APS contributions in an integrated solver” on page 5.

Reviewer 3 Report

Comments and Suggestions for Authors

Paper is very interesting and can be a good contribution on APS algorithms. I think the manuscript should be accepted, although the authors should consider the following comments.

What was the criterion for considering a temporary baseline of 120 days? Please indicate this in the text.

It would be interesting to include a table in the methodology that collects all the parameters used in the subsection 2.3.3 D-InSAR processing.

Isn't the coherence threshold value of 0.15 too low? It would be good to discuss it in the text.

In Table 1, organize and sort the data shown, as there are some that are out of order.

Indicate in the conclusions that the real data consisted of applying the algorithm in the 2020 Mw 5.7 Dingjie Earthquake.

Comments on the Quality of English Language

Be consistent throughout the text with abbreviations and acronyms. Either all uppercase or all lowercase. In my opinion, it should always be in uppercase. For Example, in Introduction: "Persistent Scatterer Interferometry (PS-InSAR) [8,31], Small Baseline Subset (SBAS) [32-34]".

Reviewer 4 Report

Comments and Suggestions for Authors

The manuscript addresses a topic of significant importance to the scientific community. However, it is evident that the paper requires substantial improvements before it can be considered for publication in its present form. Several critical issues need to be addressed:

·         Language and Clarity: The paper suffers from a lack of clarity due to poor English language usage and multiple spelling errors. To enhance readability, I recommend thorough proofreading and language editing.

·         Specific Technical Points: The authors should address the following technical issues in their revision:

a. Provide the RMS error of the estimated APS for simulated data, including variations with (a) the number of images, (b) the number of interferograms, (c) non-linearity of displacement, and (d) phase noise. Compare results between the "standard CSS algorithm" from [36] and the "enhanced CSS algorithm" (ECSS) developed by the authors. Show numerical values.

b. Correct the erroneous reference to the average tropospheric signal over Mexico, as it pertains to the average improvement obtained by the authors of [22] and not directly to the region (0.45cm/km).

c. Clarify the expression "atmospheric or dem datasets" in the Introduction.

d. Include the complete equation (2) from [36] in the manuscript.

e. Define the Atmospheric Noise Coefficient (ANC).

f. Explain how equation (5) changes when introducing model (6).

g. Describe the process of simulating APS for each SAR image.

h. Evaluate the necessity of equations (8), (9), and (10), and provide a clear explanation of the "smoothing factor" set to "100."

i. Improve the clarity of Table 1 and Figure 5 for better comprehension.

m. some bibliographic references are not useful

In summary, the paper's content requires a comprehensive review and revision to enhance its clarity and scientific merit. I strongly recommend that the authors address these issues to increase the paper's chances of being considered for publication.

Comments on the Quality of English Language

The English language is quite poor and many spelling errors are present.

Round 2

Reviewer 1 Report

Comments and Suggestions for Authors

I have no more comment for this version of the manuscript.

Comments on the Quality of English Language

English language is fine.

Reviewer 2 Report

Comments and Suggestions for Authors

No further changes are required.

Reviewer 4 Report

Comments and Suggestions for Authors

The manuscript has been significantly improved. I would simply recommend the authors to define more clearly the "recovery ratio".
